# Hiss and tell: What influences venom yields of India's big four snakes?

Prasad Gopalkrishna Gond☯, Mihir Kumar☯, Ajinkya Unawane☯, Kartik Sunagar ⓘ *

Evolutionary Venomics Lab, Centre for Ecological Sciences, Indian Institute of Science, Bangalore, Karnataka, India

☯ These authors contributed equally to this work.
* ksunagar@iisc.ac.in

## Abstract

Snake venoms have evolved over millions of years to subdue prey and deter predators rapidly. The potency and amount of venom differ both within and across species, and are primarily influenced by their ecology and environment. Understanding venom yields in clinically relevant species is essential for refining treatment approaches for snakebite. Despite this, venom yields have seldom been documented, especially in snakes of the Indian subcontinent. To address this knowledge gap, we analysed venom yields from 338 specimens of the clinically most important "big four" Indian snakes—*Naja naja*, *Daboia russelii*, *Bungarus caeruleus*, and *Echis carinatus*—collected across diverse biogeographic and agroclimatic zones. We systematically compared yields across different genders, regions, and life stages to uncover patterns in venom production and explored the relationship between venom yield and dry weight. Our findings reveal substantial inter- and intraspecific variation, with *N. naja* and *D. russelii* exhibiting the highest average yields, while *B. caeruleus* and *E. carinatus* produced much lower quantities. Geographic variation was significant for *B. caeruleus*, but not for the other species. No sex-based differences were observed; however, life stage was an important determinant, with adults producing more venom than juveniles or subadults. Comparative analysis with captive populations indicated that captivity may not substantially alter venom yields. By integrating venom yield and toxicity data, we highlight the distinct envenomation strategies among the big four snakes and their implications for snakebites. These insights are crucial for improving antivenom production, clinical management, and understanding the ecological and evolutionary dynamics of Indian snake venoms.

## Author summary

Snakebite is a neglected tropical disease causing substantial mortality and disability in India. Although venom composition and toxicity are well characterised

**Data availability statement:** All relevant data are available within the manuscript and its supplementary files.

**Funding:** KS was supported by the DBT/Wellcome Trust India Alliance Fellowship [IA/I/19/2/504647]. This work was also supported by the Department for International Development (IAVI/BES/KASU/0002), the DST-INSPIRE Faculty Award (DST/INSPIRE/04/2017/000071), and Wellcome Trust grant (223619/Z/21/Z) to KS. The funders had no role in study design, data collection and analysis, decision to publish, or preparation of the manuscript.

**Competing interests:** The authors declare that there are no competing interests.

for the "big four" medically most important snakes—*Naja naja*, *Daboia russelii*, *Bungarus caeruleus*, and *Echis carinatus*—venom production dynamics in wild populations remain poorly understood. We collected and quantified venom from 338 wild snakes spanning India's major bioclimatic zones. Venom yield differed markedly among species, with *D. russelii* and *N. naja* producing greater volumes than *B. caeruleus* and *E. carinatus*. Intraspecific variation was pronounced, geography and sex-related differences exerted minor effects, whereas age was the dominant predictor, with adults consistently yielding more venom than juveniles. Integrating venom yield with toxicity data reveals distinct envenomation strategies across species. These baseline data have immediate implications for antivenom manufacturing and clinical management, and provide interesting insights into the ecological determinants of venom production.

## 1. Introduction

India is home to an impressive variety of snakes, with over 350 species recorded to date [1]. Among these, nearly 60 are clinically relevant to humans, requiring clinical intervention to prevent severe symptoms associated with their bites. The so-called 'big four' snakes include the spectacled cobra (*Naja naja*), Russell's viper (*Daboia russelii*), common krait (*Bungarus caeruleus*), and saw-scaled viper (*Echis carinatus*), and are said to be responsible for most envenomations and snakebite-related morbidities and fatalities [2].

Snake venoms are intricate biochemical concoctions that have evolved to incapacitate and/or kill prey rapidly [3,4]. The potency and volume of venom produced vary considerably among snakes, and are linked to their ecological roles [5,6]. For instance, snakes that prey on fast, small prey could benefit from highly potent neurotoxins or procoagulant toxins that aid in rapid immobilisation. In contrast, species subduing larger prey could rely on larger volumes of venom to ensure effective dosing. Habitat and prey community composition could further shape venom repertoire, potency and yield. Understanding venom yields from clinically important snakes is therefore central to improving snakebite treatments, as it provides insight into the volume of venom that these snakes can potentially inject into their bite victims. At the global scale, a handful of studies reveal broad determinants of venom yield. Species- and size-dependent differences in venom yields without correlation to body mass have been documented in *Bothrops* spp., *Crotalus durissus terrificus*, and *Micrurus* spp. [7]. The effects of sex, geography, body size, relative head size, and species identity on venom yields have been noted in *Notechis scutatus* and *Pseudonaja textilis* [8]. Collectively, these investigations have highlighted the influence of ecological and morphological factors on snake venom yield [5]. Unfortunately, venom yields from wild snake populations have seldom been investigated, especially in the Indian subcontinent.

To date, only a single study has reported venom yields from three of the four 'big four' snake species in India, focusing exclusively on a captive cohort in Maharashtra,

Southwestern India [9]. Although this research observed seasonal variation in yields, the controlled conditions of captivity prevent an accurate assessment of the actual impact of abiotic factors on venom yields. To date, no study has generated comprehensive data on venom yields from wild snake populations in India.

We seek to fill this knowledge gap by analysing venom yields from over 338 wild-caught snake specimens from distinct biogeographic/agroclimatic zones across India. We systematically assess differences in venom yields based on gender, geography, and life stage, aiming to identify patterns and underlying variations. The data being reported here are vital for both medical and ecological research.

## 2. Methodology

### 2.1. Ethical statement

Samples of human origin were not utilised in this study. No experimentation on animals was conducted, and the $LD_{50}$ values have been reported previously.

Snake venoms were collected from six biogeographic regions across India with permissions from the State Forest Departments: Andhra Pradesh (Permit No: 16284/2016/WL-3, 07.02.2022), Goa (Permit No: 2-66-WL-RESEARCH PERMISSIONS-FD-2022-23-Vol.IV/858, 27.05.2022), Karnataka (Permit No: PCCF(WL)/E2/CR-06/2018-19, 23.02.2022), Kerala (Permit No: KFDHQ-1006/2021-CWW/WL10, 10.06.2022), Maharashtra Desk-22/(8)/WL/Research/CR-60(17-18)/3349/21-22, 21/03/2022), Madhya Pradesh (No./M.CH-II/Research/F-214/21932), Rajasthan (Permit No: F19(29) Permission/ CWLC/ 2017-18/ 595, 09/05/2018), Tamil Nadu (WL5(A)/33005/2017, Permission No. 61/2023), Uttarakhand (Letter No: 2519/5-6, 21.03.2024), and West Bengal (Memo No:1023/WL/4R-12/2022, 21.04.2022) India. Venoms were collected from male and female individuals of all developmental stages, including neonates, juveniles, sub-adults and adults (S1 Table). Highly experienced herpetologists identified the developmental stage, sex and species.

### 2.2. Venom extraction and storage

Between 2021 and 2024, 338 snake venom samples were collected from wild-caught snakes during rescue operations coordinated by the State Forest Department and local snake rescuers. After extracting the venom, the snakes were released back into their natural habitat under the supervision of the forest department. Experienced herpetologists employed safe handling methods, using acrylic restraining tubes to restrain snakes manually. Venom was collected by encouraging the snake to bite onto a sterile parafilm stretched over a beaker, without applying external pressure to the venom gland (S1 Fig). The extracted crude venom was promptly measured and placed in a vapour shipper for transportation. Following lyophilisation, the venom samples were stored at –80 °C until ready for further processing. The dry weight of the lyophilised venom samples was measured with a high-precision microbalance (Sartorius AG, Göttingen, Germany).

### 2.3. Statistical analysis

All statistical analyses were conducted using GraphPad Prism (version 10.0, GraphPad Software, USA; www.graphpad.com). Initially, venom yield data underwent a normality check with the Shapiro-Wilk test [10]. Since the data did not follow a normal distribution, non-parametric tests were utilised. To compare venom yields among multiple groups, the Kruskal-Wallis test was applied [11], followed by Dunn's post hoc test with Benjamini-Hochberg adjustment for multiple-comparison corrections [12]. Effect sizes for the Kruskal-Wallis tests were determined using epsilon-square (R 4.5.1; www.R-project.org). For comparisons between two groups, the Mann-Whitney U test was used [13], followed by a suitable post hoc test, with effect sizes calculated using rank-biserial correlation (r) [14]. Additionally, the Median Absolute Deviation (MAD) and the Coefficient of Variation (CoV) analyses were conducted using R. Detailed results of the normality test, along with the code for effect sizes, MAD, and CoV, are provided in S1 File.

## 3. Results

### 3.1. Venom yields from the pan-Indian populations of the 'big four' snakes

A total of 338 venom samples were collected across various biogeographic regions in India, including the coastal plains, the Deccan Peninsula/Deccan Plateau, the Gangetic Plains, semi-arid regions, the Western Ghats, and the deserts (Table 1 and Fig 1).

### 3.2. Ecological implications on snake venom yields

**3.2.1. Interspecific variation.** Venom yields showed considerable variation among the *E. carinatus* subspecies, as well as the other 'big four' species investigated in this study (Tables 2 and S1 and Fig 2A). *Bungarus caeruleus* and *E. c. carinatus* had the lowest mean yields, with *B. caeruleus* averaging 8.95 ± 1.28 mg (range: 0.10 – 32.00 mg) and *E. c. carinatus* averaging 2.76 ± 0.35 mg (range: 0.20 – 8.20 mg). *E. c. sochureki* yielded intermediate amounts, with a mean of 50.01 ± 14.38 mg (range: 9.92 – 123.20 mg). The highest mean yields were recorded for *D. russelii* (106.60 ± 9.61 mg, range: 0.50 – 749.00 mg) and *N. naja* (136.10 ± 11.16 mg, range: 0.50 – 800.40 mg).

Ranking of median venom yields mirrored the means, with *E. c. carinatus* producing the lowest median yield (2.76 mg) and *D. russelii* the highest (86.14 mg) (Tables 2 and S1 and Fig 2A). Kruskal–Wallis confirmed these findings, where *N. naja* and *D. russelii* were found to produce significantly higher amounts than *B. caeruleus* and *E. carinatus* ($\chi^2 = 139.6$, df = 4, $p < 0.0001$; Fig 2A).

Geographic comparisons between populations of the 'big four' snake species revealed statistically significant differences in venom yields (Kruskal–Wallis $\chi^2 = 12.43$, df = 5, $p = 0.02$; Fig 2B), but post hoc Dunn's tests with multiple comparison correction were insignificant (adjusted $p > 0.05$).

Similarly, although a statistically significant overall difference in venom yield was observed between males and females of the 'big four' Indian snakes (Mann-Whitney U = 6256, $p = 0.0325$; Fig 2C), sex-specific differences were not found within any species under investigation (adjusted $p > 0.05$).

In contrast, ontogenetic assessment revealed that venom yield increases substantially from the juvenile to the adult stage (Kruskal–Wallis $\chi^2 = 44.38$, df = 3, $p < 0.0001$; Fig 2D), highlighting life stage as a key factor influencing venom production.

**3.2.2. Intraspecific variation. Biogeographic variation:** Venom yield in *N. naja* (n = 144; mean = 136.10 mg, median = 83.64 mg, range: 0.50–800.40 mg; S1 Table) did not differ significantly across the six biogeographic zones of India (Kruskal–Wallis $\chi^2 = 3.103$, df = 5, $p = 0.684$; Fig 3A). Although the highest median yield was observed in the semi-arid zone (194.00 mg) and the lowest in the Western Ghats (73.00 mg), these differences were not statistically significant ($p = 0.68$).

**Table 1. Details of venom samples of *N. naja*, *D. russelii*, *B. caeruleus,* and both subspecies of *Echis*, including the sampling region and the number of samples collected, are provided.**

| Species | Biogeographic zone | | | | | | Total |
|---|---|---|---|---|---|---|---|
| | Coast | Deccan Peninsula | Gangetic Plains | Semi-arid | Western Ghats | Desert | |
| *B. caeruleus* | 3 | 14 | 8 | NA | 16 | NA | 41 |
| *N. naja* | 21 | 55 | 11 | 3 | 51 | 3 | 144 |
| *D. russelii* | 3 | 54 | 14 | 2 | 42 | NA | 115 |
| *E. c. carinatus* | 12 | 8 | NA | NA | 10 | NA | 30 |
| *E. c. sochureki* | NA | NA | NA | NA | NA | 8 | 8 |
| Total | 39 | 131 | 33 | 5 | 119 | 11 | 338 |

NA denotes areas where the respective snake species or subspecies is undocumented.

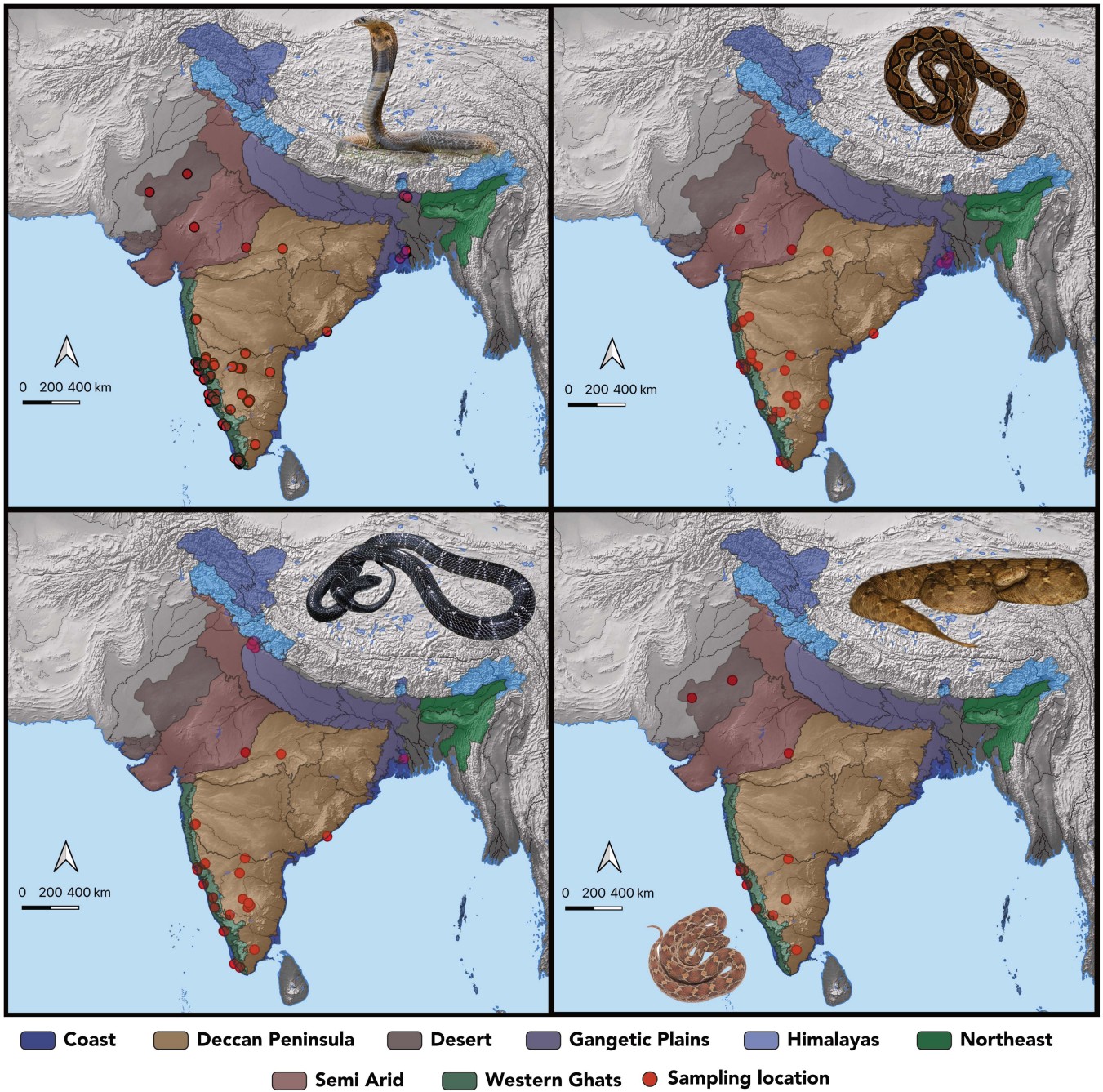

**Fig 1. Sampling locations of the 'big four' snakes across India.** Venom collection sites (red spheres) are marked on a map of India with uniquely colour-coded biogeographic regions. Representative images of the 'big four' snakes and *E. c. sochureki* are also shown in their respective panels. The maps were generated using QGIS v3.42 (www.qgis.org; vector map: Natural Earth: www.naturalearthdata.com/about/terms-of-use). Photo credits: Kartik Sunagar, Ashok Captain, and Ajinkya Unawane.

**Table 2. Venom yields of the 'big four' snakes, including both subspecies of *Echis* in India.**

| Species | N | Venom yield (mg) | | |
|---|---|---|---|---|
| | | Mean±SEM | Median | SD |
| *N. naja* | 144 | 136.10±11.16 | 83.64 | 133.90 |
| *D. russelii* | 115 | 106.60±9.61 | 86.14 | 103.10 |
| *B. caeruleus* | 41 | 8.95±1.28 | 6.00 | 8.18 |
| *E. c. carinatus* | 30 | 2.76±0.35 | 2.76 | 1.91 |
| *E. c. sochureki* | 8 | 50.01±14.38 | 38.54 | 40.69 |

N=Number of snakes; SEM=Standard Error of the Mean; SD=Standard Deviation.

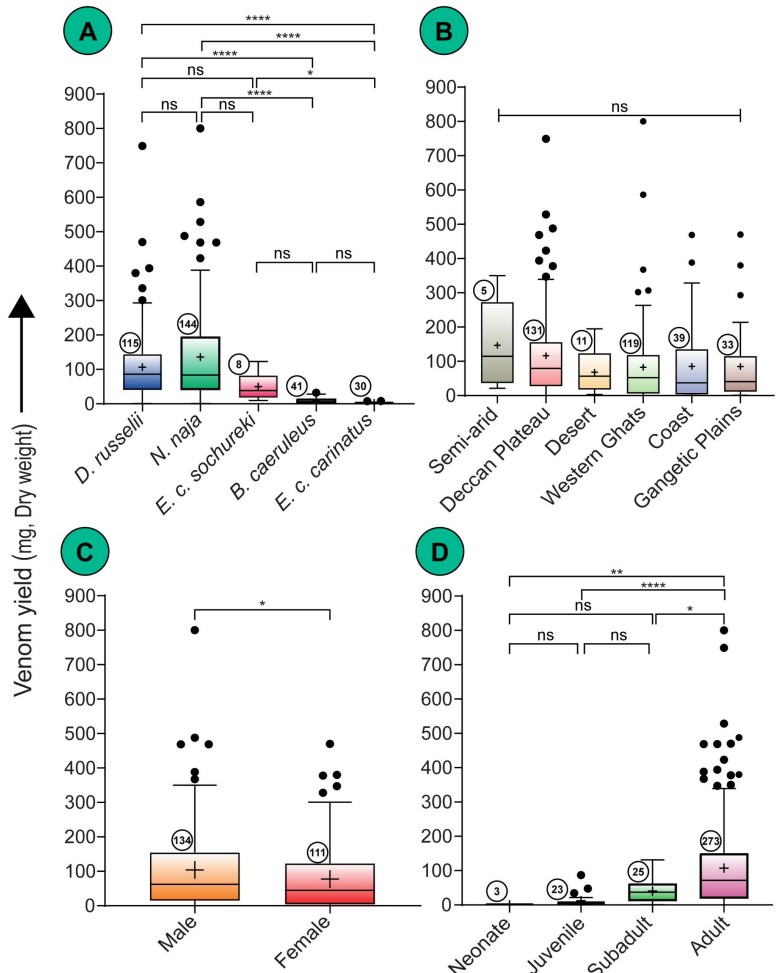

**Fig 2. Patterns of venom yield in the 'big four' snakes.** (A) among species, (B) among biogeographic regions, (C) between the two sexes, and (D) throughout ontogenetic stages. Data are represented as Tukey boxplots showing median (represented by a line inside the box), interquartile ranges (represented by the box), 1.5 x the interquartile range (represented by whiskers), and outliers (represented by dots). Statistical comparisons were performed using non-parametric tests. Here, *: $p < 0.05$; **: $p < 0.01$; ****:$p < 0.0001$; ns: non-significant;+: mean. Numbers within circles represent sample size.

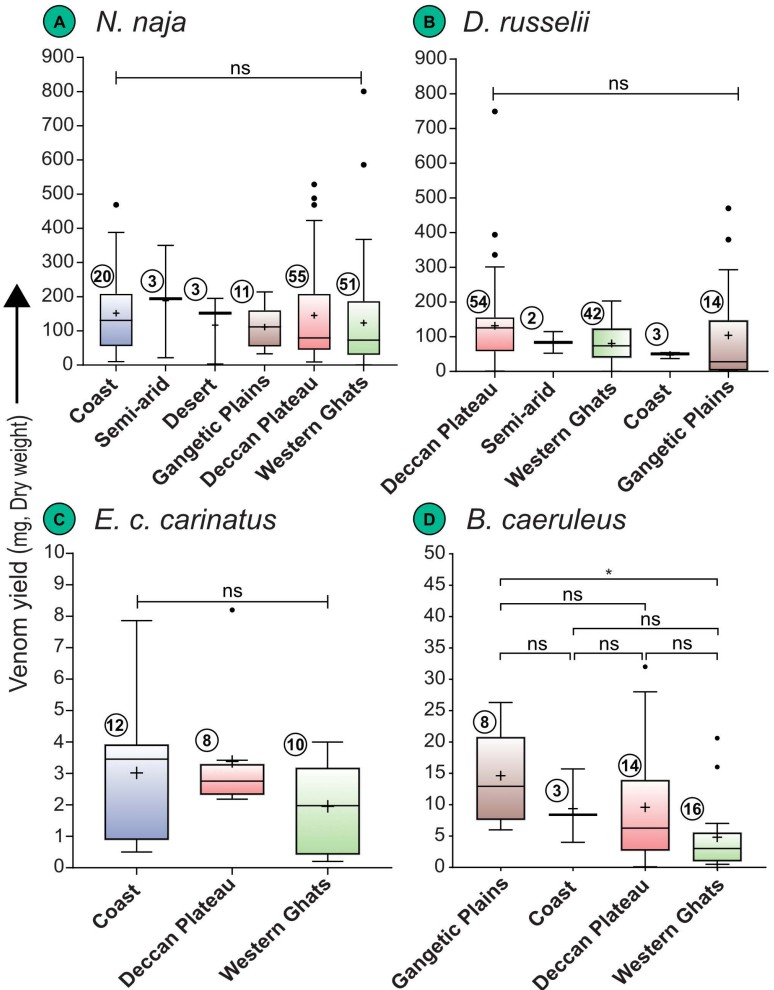

**Fig 3. Biogeographic variation in venom yields of the 'big four' Indian snakes.** (A) *N. naja*, (B) *D. russelii*, (C) *E. c. carinatus*, and (D) *B. caeruleus*. Data are represented as Tukey boxplots, showing the median (represented by a line inside the box), interquartile ranges (represented by the box), 1.5 x the interquartile range (represented by whiskers), and outliers (represented by dots). Statistical comparisons were performed using non-parametric tests. Here, ***: p < 0.001; ****: p < 0.0001; ns: non-significant; +: mean. Numbers within circles represent sample size.

Similarly, venom yield in *D. russelii* (n = 115; mean = 106.60 mg, median = 86.14 mg, range: 0.50–749.00 mg) varied significantly across India's biogeographic zones (Kruskal–Wallis $\chi^2$ = 13.05, df = 4, p = 0.011; Fig 3B), but post hoc Dunn's tests with multiple comparison correction did not find significant pairwise differences (adjusted p > 0.05). The highest median yield was recorded in the Deccan Peninsula (125.50 mg), while the Gangetic Plains had the lowest (28.00 mg).

Venom yield in *E. c. carinatus* (n = 30; mean = 2.76 mg, median = 2.75 mg, range: 0.20–8.20 mg) did not vary significantly across the Coast, Deccan Peninsula, and Western Ghats biogeographic zones (Kruskal–Wallis $\chi^2$ = 2.175, df = 2, p = 0.337; Fig 3C), with median yields consistently ranging from 1.98 to 3.46 mg.

Unlike the other big four species, venom yield in *B. caeruleus* (n = 41; mean = 8.95 mg, median = 6.00 mg, range: 0.10–32.00 mg) showed significant geographic variation (Kruskal–Wallis $\chi^2$ = 7.824, df = 3, p = 0.0498; Fig 3D). Dunn's post-hoc analysis identified significant differences between the Gangetic Plains and Western Ghats (Adjusted p = 0.0051), with the highest median yield observed in the Gangetic Plains (12.94 mg) and the lowest in the Western Ghats (4.00 mg).

**Sex-specific differences:** Comparisons between sexes in *N. naja* revealed no significant variation in venom yield (Mann-Whitney U = 858.5, p = 0.254; Fig 4A). Males had a mean venom yield of 141.00 mg, while females averaged 114.00 mg, with considerable overlap in their datapoints.

Similarly, there were no significant differences in venom yield between male and female *D. russelii* (Mann-Whitney U = 727.5, p = 0.383; Fig 4B), with females (n = 48, mean = 107.90 mg, median = 102.00 mg) and males (n = 27, mean = 85.32 mg, median = 70.00 mg) showing considerable overlap in their datapoints.

Sex-specific analysis in *E. c. carinatus* did not show significant differences in venom yields either (Mann-Whitney U = 97, p = 0.946; Fig 4C), as males (mean = 2.77 mg, median = 2.71 mg) and females (mean = 2.74 mg, median = 2.71 mg) had nearly identical values.

Statistically insignificant differences in venom yield were documented between the males and females of *B. caeruleus* (Mann-Whitney U = 75, p = 0.479; Fig 4D). Male (mean = 9.62 mg, median = 6.50 mg) and female (mean = 8.63 mg, median = 5.13 mg) yields overlapped substantially.

**Ontogeny-specific differences:** Unlike biogeographic and sex-specific variation, analysis by life stage in *N. naja* showed statistically significant differences (Kruskal–Wallis χ² = 39.804, df = 2, p < 0.0001). Adults produced the highest

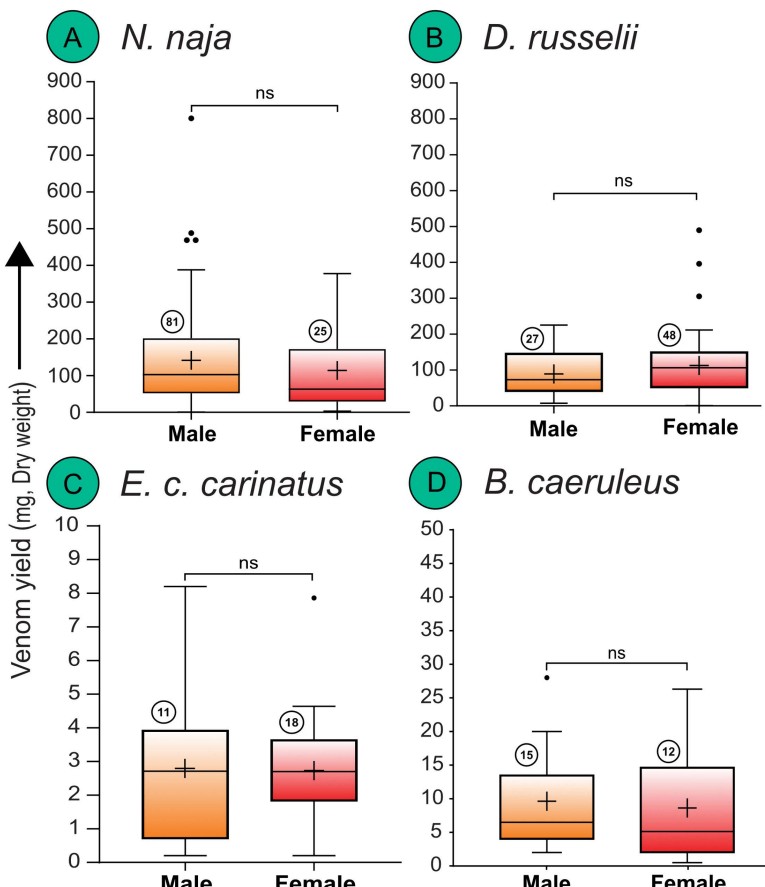

**Fig 4. Sex-specific variation in venom yields of the 'big four' Indian snakes.** (A) *N. naja*, (B) *D. russelii*, (C) *E. c. carinatus*, and (D) *B. caeruleus*. Data are represented as Tukey boxplots showing median (represented by a line inside the box), interquartile ranges (represented by the box), 1.5 x the interquartile range (represented by whiskers), and outliers (represented by dots). Statistical comparisons were performed using non-parametric tests. Here, ns: non-significant; +: mean. Numbers within circles represent sample size.

median venom yield at 125.00 mg, whereas subadults and juveniles had much lower median yields of 47.60 mg and 6.50 mg, respectively (Fig 5A and S1 Table).

Similarly, analysis by life stage in *D. russelii* indicated significant differences in venom yield (Kruskal–Wallis $\chi^2 = 27.777$, df = 2, p < 0.001). Adults had the highest median yield at 95.69 mg, subadults showed intermediate yields (median = 33.22 mg), and juveniles had the lowest yields (median = 3.00 mg; Fig 5B).

Ontogenetic comparisons in *E. c. carinatus,* too, revealed highly significant differences (Kruskal–Wallis $\chi^2 = 11.72$, df = 3, p < 0.0028). Adults produced the highest median venom yield at 3.00 mg, while subadults and neonates had substantially lower yields, with medians below 1 mg (Fig 5C and S1 Table).

Ontogenetic analysis was not conducted for *B. caeruleus* due to the limited number of venom samples from the juvenile and subadult specimens.

**3.2.3. Interindividual variation in venom yields of the 'big four' snakes.** Interindividual variability in venom yield was evident across all 'big four' snake species (Fig 6). This variation was quantitatively assessed using the MAD (Fig 6A) and CoV (Fig 6B). Interindividual variation in venom yield was more pronounced in *N. naja, D. russelii,* and *E. c. sochureki* compared to *B. caeruleus* and *E. c. carinatus*.

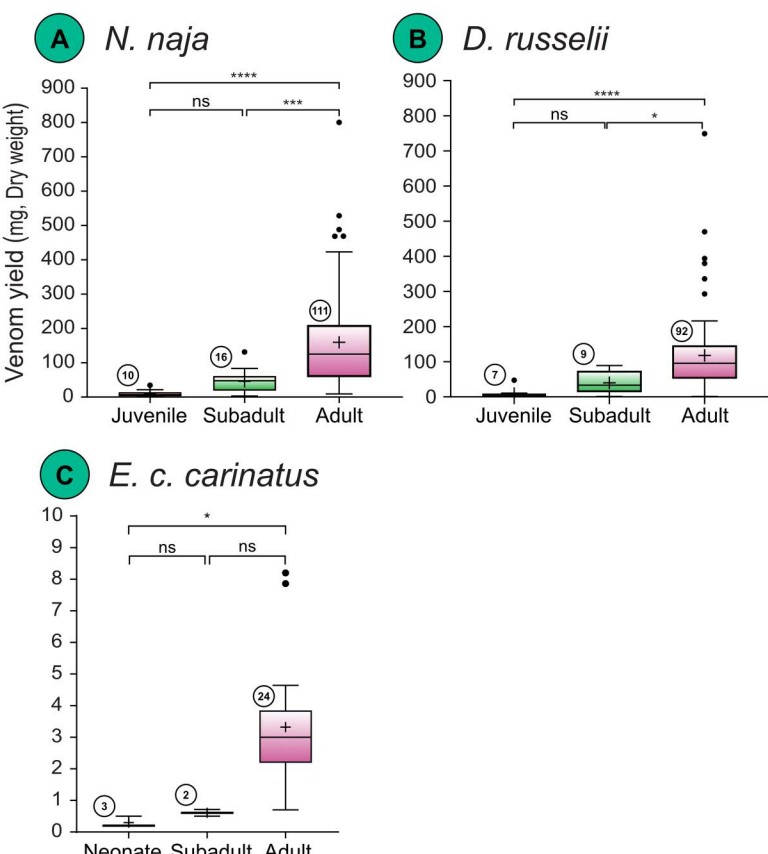

**Fig 5. Ontogenetic variation in venom yields of the 'big four' Indian snakes.** (A) *N. naja,* (B) *D. russelii* and (C) *E. c. carinatus*. Data are represented as Tukey boxplots showing median (represented by a line inside the box), interquartile ranges (represented by the box), 1.5 x the interquartile range (represented by whiskers), and outliers (represented by dots). Statistical comparisons were performed using non-parametric tests. Here, *: p < 0.05; ***: p < 0.001; ****: p < 0.0001; ns: non-significant; +: mean. Numbers within circles represent sample size.

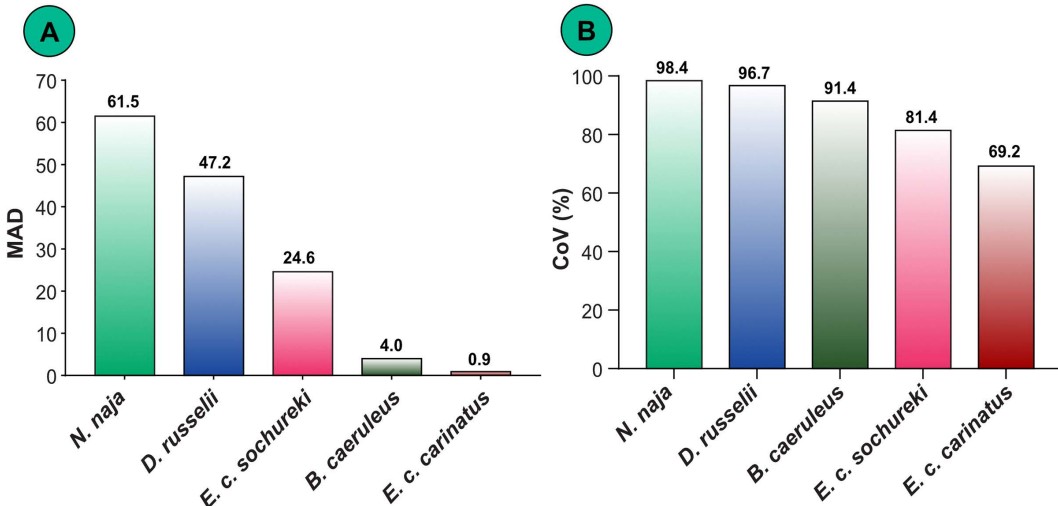

**Fig 6. Interindividual differences in venom yields.** (A) Median absolute deviation (MAD), and (B) Coefficient of Variation (CoV) in venom yields across individuals of the 'big four' snakes.

## 4. Discussion

Our comprehensive and biogeographically inclusive assessment of venom yield in the pan-Indian populations of 'big four' snakes provides the first detailed insights into how ontogeny, sex, and geography affect venom yield in wild populations of India's 'big four' snakes.

Snakes were captured from the wild for sample collection and then safely released back at their original locations. This approach proved highly effective and non-harmful, as demonstrated by the recapture of some of the previously marked *E. c. carinatus* individuals in the same area in subsequent years. Our analyses revealed that venom yields varied significantly among the 'big four' species, with *N. naja* (mean = 136.10 ± 11.16 mg) and *D. russelii* (mean = 106.60 ± 9.61 mg) producing the largest quantities, while *B. caeruleus* (mean = 8.95 ± 1.28 mg) and *E. carinatus* (mean = 2.76 ± 0.35 mg) yielded lower amounts of venom (Tables 2 and S1). This pattern likely reflects the influence of body size, as larger snakes had higher venom yields than smaller species (Fig 2A)—a trend supported by previous research showing a positive correlation between body size and venom yield [15–18].

Biogeographic variation in venom yield was evident for *B. caeruleus,* with individuals from the Gangetic Plains producing substantially higher venom yields compared to those from the Western Ghats (adjusted p = 0.0051; Fig 3D). This pattern may reflect ecological adaptation to local environmental pressures, such as geographic differences in prey availability, diet breadth, and habitat structure. In contrast, *N. naja, D. russelii*, and *E. c. carinatus* did not exhibit significant biogeographic differences in venom yield (Fig 3A–C). Notably, sexual dimorphism did not appear to influence the venom yield across species (Fig 4A–D), which contrasts with some earlier studies that reported higher yields in females [15]. The datapoints of venom yield for males and females of all four species overlapped considerably (S1 Table).

Ontogenetic shifts in venom composition, function, and potency have been reported for *D. russelii* in India [19]. A similar developmental shift in venom composition was also observed in *N. naja*, but it did not appear to affect the potency of the venom [19]. Ontogenetic shifts remain unstudied for the other two members of the 'big four' snakes. Ontogenetic analysis demonstrated that life stage is a primary determinant of venom output. Adults produced the highest median yields, while subadults and juveniles yielded lower amounts (Fig 5A–C and S1 Table). For example, adult *N. naja* had a median yield of 125.00 mg, compared to 47.60 mg for subadults and 6.50 mg for juveniles. Similar trends were observed in *D. russelii*

and *E. c. carinatus*. Body size is a key determinant of venom yield and may confound comparisons across life stages. Because morphometric data, such as length and body mass, were not consistently available for wild-caught specimens, we classified life stage using size proxies and expert field assessment—an approach that is practical but necessarily coarse. Even with this limitation, the large sample size and consistent species-level categorisation support the conclusion that life stage is a primary driver of variation in venom yield. However, body-size effects cannot be fully disentangled. We did not perform an ontogenetic analysis for *B. caeruleus* due to the insufficient number of samples from juvenile and adult stages to support a robust comparison. Collecting krait venom from wild populations is logistically challenging given their elusive behaviour and low encounter rates [20].

We also compared venom yields from wild-caught snakes to previously published data from captive populations [9,21]. No significant differences were found between wild and captive yields (S2 Fig), suggesting that captivity may not substantially alter venom yields under artificial conditions. However, this contrasts with previous research, which has shown that the first milking session in captivity often yields more venom than subsequent extractions, and that factors, such as health, feeding, and extraction intervals, can influence output [22]. Previous work also reported seasonal variation in venom yields in captive snakes [23]. However, captivity constrains environmental exposure, limiting our ability to infer about abiotic drivers. In this study, seasonality could not be evaluated robustly because sampling was opportunistic and dictated by forest department regulations and permit restrictions, species availability, and field logistics, resulting in insufficiently balanced seasonal sample sizes for statistical analysis. We acknowledge this limitation and highlight the need for future, seasonally structured field sampling to quantify the effects of seasonality on venom yield.

$LD_{50}$ values and venom yield data were analysed together to illustrate the distinct envenomation strategies among the big four snakes (Table 3). *Bungarus caeruleus* stands out for its exceptional potency ($LD_{50} = 0.11$ mg/kg) despite having one of the lowest average venom yields (8.95 mg). In contrast, *N. naja* exhibited moderate toxicity ($LD_{50} = 0.42$ mg/kg) with the highest average venom yield (136.10 mg), which may contribute to the higher rates of snakebite mortality associated with this species. *D. russelii* produced very high venom yields (106.60 mg) and a comparatively low $LD_{50}$ (0.31 mg/kg), while *E. c. carinatus* has the least toxic venom ($LD_{50} = 0.64$ mg/kg) and the smallest venom yield (2.76 mg), yet remains a leading cause of envenomation. These findings suggest that snakebite lethality is not determined solely by the quantity of venom, but also by the venom composition and pathophysiological targets. For example, the venoms of *B. caeruleus* and

**Table 3. The median lethal dose of the 'big four' Indian snakes.**

| Species | Biogeographical Zones | | | | | Average $LD_{50}$ | Average yield (mg) |
|---|---|---|---|---|---|---|---|
| | Deccan Peninsula | Semi-arid | Western Ghats | Coast | Gangatic Plains | | |
| *Naja naja* | 0.22 | 0.33 | 0.73 | 0.55 | 0.27 | 0.42 | 136.10 |
| | [25,32] | | | | | | |
| *Daboia russelii* | 0.166 | 0.14 | 0.19 | 0.18 | 0.34 | 0.20 | 106.60 |
| | [19,29] | | | | | | |
| *Bungarus caeruleus* | 0.125 | 0.143 | 0.13 | 0.09 | 0.069 | 0.11 | 8.95 |
| | [20] | | | | | | |
| *Echis c. carinatus* | 0.55 | NA | 0.61 | 0.753 | NA | 0.64 | 2.76 |
| | [33] | | | | | | |
| *Echis c. sochureki* | 1.76 | NA | NA | NA | NA | 1.76 | 50.01 |
| | [32] | | | | | | |

This table summarises the median lethal dose ($LD_{50}$; mg/kg) of venoms from the 'big four' Indian snakes, including both subspecies of *Echis,* across biogeographies. The $LD_{50}$ values (male CD-1 mice; 18–22 g) are derived from previous studies, with references indicating the primary sources from which these values were obtained.

*N. naja* are dominated by neurotoxins that rapidly block neuromuscular transmission [19,24–27]. In contrast, *D. russelii* and *E. carinatus* produce venom rich in proteolytic enzymes [28–31]. Moreover, the clinical outcome may also depend on several other factors, including the injected dose, bite site, victim size and health, and treatment timeliness. Together, these findings indicate that the snakebite burden is shaped not only by venom potency alone, but also by the interplay of potency, venom yield, and other factors.

Overall, our research fills a critical gap in the understanding of venom yield variation in Indian snakes, highlighting the importance of size, geography, and life stage, while finding no evidence for the influence of sex-related differences. These findings have important implications for antivenom production, clinical management of snakebite, and ecological and venomics studies. For instance, the data generated here can guide the formulation of immunisation mixtures for antivenom production and inform potency specifications when integrated with toxicity data. As a result, this could greatly enhance neutralisation potencies against doses typically injected by these snakes. Clinically, species- and biogeographic region-specific yield ranges can help refine antivenom dosing algorithms, thereby improving treatment outcomes. This data can also guide product stock planning, aligning antivenom supply with expected envenomation burden. Moreover, the absence of stark differences in venom yields between wild and captive cohorts suggests that captive snakes can be viable sources for antivenom production, provided that geographic and ontogenetic diversity is well-represented. From an evolutionary ecological standpoint, the data reveal distinct envenomation strategies—high yield/moderate toxicity (e.g., *N. naja*, *D. russelii*) versus low yield/high venom potency (e.g., *B. caeruleus*)—, which could inspire focused studies on diet, habitat, and physiology as drivers of venom production. By providing robust, regionally representative data, the study supports efforts to improve snakebite treatment and deepen knowledge of venom biology in India.

## Supporting information

**S1 File. R script and normality-test results.** This file provides the complete R code used for analysing venom yield data, including the Shapiro-Wilk normality test, effect-size calculations, median absolute deviation (MAD), and coefficient of variation (CoV).
(DOCX)

**S1 Fig. Snake venom extraction.** The image depicts the *Naja naja* venom extraction procedure, in which the snake is gently coaxed to bite a membrane stretched over a sterile beaker, allowing for the collection of venom without the need for external pressure. Photo credit: Ajinkya Unawane.
(DOCX)

**S2 Fig. Comparison of venom yields between captive and wild snakes.** In the figure, venom yield (measured in mg, dry weight) was compared between previously published data and the findings from the current study. For *N. naja*, *D. russelii*, and *B. caeruleus* (panel A), as well as *E. c. carinatus* and *E. c. sochureki* (panel B), venom yields from captive snakes, as reported by Whitaker and Whitaker (2012) and Tumbare and Khadilkar (2004), were assessed alongside the venom yields collected from wild snakes in this study. The data are expressed as mean±SEM (represented by error bars), with non-significant comparisons indicated by "ns".
(DOCX)

**S1 Table. Venom yield statistics for the 'big four' Indian snakes.** This table summarises venom yield statistics for the 'big four' Indian snakes: *N. naja, D. russelii, B. caeruleus,* and *E. c. carinatus* (including its subspecies *E. c. sochureki*). The data are presented as sample size (N), mean±SEM (standard error of the mean), median, standard deviation (SD), minimum venom yield (min), first quartile (Q1), third quartile (Q3), and maximum venom yield (max). For some samples, locality, gender, and/or age were not recorded.
(DOCX)

## Acknowledgments

The authors are thankful to the following State Forest Departments for the kind support and permits for venom collection: Andhra Pradesh, Goa, Karnaka, Kerala, Maharashtra, Madhya Pradesh, Rajasthan, Tamil Nadu, Uttarakhand and West Bengal. The authors are thankful to Ajay Kartik and Gerry Martin for their invaluable assistance in collecting some of the samples used in this study. The authors are grateful to Anwar Yusuf (ACF, Kerala FD), Akash Verma (IFS, Uttarakhand FD), and Gnaneshwar CH for their support during venom collection. We would like to thank the following snake rescuers for assistance with sample collection: Vivek Sharma, Amrit Singh, Charan Desai, Mirjoy Mathew, Abhijeet Patil, Swanand Patil; Sachin Gowda, Anand Chitti, Shivani Chitti, Susmita Nesarikar, Jeet Roche, Bhuwan Devadiga, Afsar Hussain, Aditya Vattam, Amit Tapase, Kushal Kori, Ajith, Omkar Pai, Vinayak Bhatt, Suhas Hegade, Pavan Naik, Afran, Poorna, Akshay Sheth, Sammilan Shetty; Javad Kudukkan; Vivek Rathod, Kiran Kumar, and Sabbavarapu Madhu; Shankar Wadkar, Pratik Mahamuni, Deepak Pacharne; Vishwa; Vishal Varma, Shyam Govindsar, and Padma Singh Rathore; Sudarshan Singh, and Arshad Khan; and Suvrajyoti Chatterjee, Nandu Kumar.

## Author contributions

**Conceptualization:** Kartik Sunagar.

**Data curation:** Prasad Gopalkrishna Gond, Ajinkya Unawane.

**Formal analysis:** Prasad Gopalkrishna Gond, Mihir Kumar, Ajinkya Unawane, Kartik Sunagar.

**Funding acquisition:** Kartik Sunagar.

**Investigation:** Prasad Gopalkrishna Gond, Kartik Sunagar.

**Methodology:** Prasad Gopalkrishna Gond, Mihir Kumar, Ajinkya Unawane.

**Resources:** Kartik Sunagar.

**Supervision:** Kartik Sunagar.

**Validation:** Prasad Gopalkrishna Gond.

**Visualization:** Prasad Gopalkrishna Gond, Mihir Kumar, Kartik Sunagar.

**Writing – original draft:** Prasad Gopalkrishna Gond, Mihir Kumar, Ajinkya Unawane, Kartik Sunagar.

**Writing – review & editing:** Prasad Gopalkrishna Gond, Mihir Kumar, Ajinkya Unawane, Kartik Sunagar.

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
