## [Decision Letter · Decision Letter 0]

21 Sep 2025

Hiss and tell: What influences venom yields of India's big four snakes?

Dear Dr. Sunagar,

Thank you for submitting your manuscript to PLOS Neglected Tropical Diseases. After careful consideration, we feel that it has merit but does not fully meet PLOS Neglected Tropical Diseases's publication criteria as it currently stands. Therefore, we invite you to submit a revised version of the manuscript that addresses the points raised during the review process.

Please submit your revised manuscript within 60 days Nov 20 2025 11:59PM. If you will need more time than this to complete your revisions, please reply to this message or contact the journal office at plosntds@plos.org. Please include the following items when submitting your revised manuscript:

We look forward to receiving your revised manuscript.

Kind regards,

Wuelton Monteiro, Ph.D.

Section Editor

Wuelton Monteiro

Section Editor

Shaden Kamhawi

co-Editor-in-Chief

Paul Brindley

co-Editor-in-Chief

**Additional Editor Comments:**

Reviewer #1:

Reviewer #2:

Reviewer #3:

Reviewer #4:

Reviewer #5:

**Journal Requirements:**

Potential Copyright Issues:

i) Please confirm (a) that you are the photographer of S1, and 2B, or (b) provide written permission from the photographer to publish the photo(s) under our CC BY 4.0 license.

ii) Figure 1A. Please (a) provide a direct link to the base layer of the map (i.e., the country or region border shape) and ensure this is also included in the figure legend; and (b) provide a link to the terms of use / license information for the base layer image or shapefile. We cannot publish proprietary or copyrighted maps (e.g. Google Maps, Mapquest) and the terms of use for your map base layer must be compatible with our CC BY 4.0 license.

If you did not receive any funding for this study, please simply state: u201cThe authors received no specific funding for this work.u201d.

7)  Please ensure that the funders and grant numbers match between the Financial Disclosure field and the Funding Information tab in your submission form. Note that the funders must be provided in the same order in both places as well. 

8) Kindly revise your competing statement to align with the journal's style guidelines: 'The authors declare that there are no competing interests.'

**Reviewers' Comments:**

Reviewer's Responses to Questions

**Key Review Criteria Required for Acceptance?**

**Methods**

-Are the objectives of the study clearly articulated with a clear testable hypothesis stated?

-Is the study design appropriate to address the stated objectives?

-Is the population clearly described and appropriate for the hypothesis being tested?

-Is the sample size sufficient to ensure adequate power to address the hypothesis being tested?

-Were correct statistical analysis used to support conclusions?

-Are there concerns about ethical or regulatory requirements being met?

Reviewer #1: (No Response)

Reviewer #2: Materials and methods

Seasonality and environmental covariates: In this study, years of collection (2021–2024) are available but not analyzed. Adding month/season to the model would visualize seasonal yield distributions by species. This could reveal phenology-linked dynamics of venom yield.

Reviewer #3: - Although the objective is clearly stated, the methodology must be complemented with, at least, the HPLC/electrophoretic venoms profile, biochemical, enzymatic and in-vivo venoms analysis (proteomics also may be useful). The venom yielded by itself (and alone) is not worthy to be published as a full research article.

- No, the study lacks of scientific support.

- Yes.

- Yes

- Yes.

- No.

Reviewer #4: Yes to most of the questions above. Please see comments to the authors for details.

Reviewer #5: (No Response)

**Results**

-Does the analysis presented match the analysis plan?

-Are the results clearly and completely presented?

-Are the figures (Tables, Images) of sufficient quality for clarity?

Reviewer #1: (No Response)

Reviewer #2: Statistical reporting for the Kruskal-Wallis test on D. russelli geographic variation (p=0.011), the post-hoc test found no significant pairs. This should be briefly discussed in the results text as it suggests that while overall variation exists, specific regional pairwise differences are not stark enough to show significance at multiple-testing correction.

Reviewer #3: -

- No.

- No.

1. Figure 1A. Please offer details about the approximately collecting areas. Where was collected each snake? India is such a huge and geographically diverse country and each of the showed biogeographic may include different ecogeographic compositions. Additionally, use colors to indicate which species was collected over each biogeographic area.

2. On table 1, “NA” means no specimens were collected? If so, please just add 0 (zero) in the table. The letters NA are not suitable for quantification purposes.

3. Table 2 vs table 1 shows a difference on the number of D. russelii and B. caeruleus individuals collected and analyzed. Please correct.

4. On table 2 please decide to use one or two significative figures. After deciding, unify with the text.

5. On figure 4, “A” does not refer to “total yield” as stated in the figure foot note. It corresponds to the variation among the geographical region. B and C also do not correspond to what is shown in the figure. Finally, there is not figure D. Please correct.

6. Figure 2B is not mentioned in the text. It must be.

7. The text does not follow an order mentioning the figures. It goes from figure 2A to figure 4A. What happened (there is not mention of) with figures 2B, 2D, 3A, 3B and 3C? These kinds of gaps are not allowed according to the manuscript instructions of the PNTD (“Place figure captions in the manuscript text in read order”). https://journals.plos.org/plosntds/s/figures

8. The same as above is observed, in the same paragraph, jumping from figure 4A to figure 4B, the go back to figure 3A, move forward to figure 6A and finally go back to figure 2C and D. All in the same paragraph.

Reviewer #4: Yes to most of the questions above. Please see comments to the authors for details.

Reviewer #5: (No Response)

**Conclusions**

-Are the conclusions supported by the data presented?

-Are the limitations of analysis clearly described?

-Do the authors discuss how these data can be helpful to advance our understanding of the topic under study?

-Is public health relevance addressed?

Reviewer #1: (No Response)

Reviewer #2: This manuscript represents important study in clinical toxinology. The data is robust, the analysis is sound, and the conclusions are well-supported and highly significant. The authors have provided a vital resource for improving snakebite management in India. The minor suggestions above are intended to further polish an already good manuscript.

Reviewer #3: The manuscript does not present conclusions, so I will refer here to the discussion section.

- No.

- No.

- Si.

1. In the discussion section, statements should be supported with the research data.

2. In the discussion section the authors are refereeing to results not mentioned or obtained in their research.

Reviewer #4: Yes to most of the questions above. Please see comments to the authors for details.

Reviewer #5: (No Response)

**Editorial and Data Presentation Modifications?**

Reviewer #1: (No Response)

Reviewer #2: (No Response)

Reviewer #3: The authors present important and interesting information about the venom yielded by the big four from different regions emphasizing on the geographic, sex, age (life stage) and size. These results are particularly important since the venoms was obtained without applying external pressure. Unfortunately, the authors present only one result which should be complemented with other information about the venoms obtained. Please find below the major findings that doesn’t allows me to recommend these results for publication at the PNTD.

Major findings.

1. In the methodology section, the authors should clearly indicate the parameters used to classify the individuals as neonate, juvenile, subadult and adult.

2. Results must be complemented with, at least, the HPLC/electrophoretic profile, biochemical, enzymatic and in-vivo analysis (proteomics also may be useful). The venom yielded by itself (and alone) is not worthy to be publisehd as a full research article.

3. The standard deviation (SD) is not a suitable parameter to indicate intraspecific variability.

4. In the discussion section, statements should be supported with the research data.

5. In the discussion section the authors are refereeing to results not mentioned or obtained in their research.

Reviewer #4: (No Response)

Reviewer #5: (No Response)

**Summary and General Comments**

Reviewer #1: The authors have presented a comparative study on venom yields and potency from the Indian “big four” snakes. Generally, the study provides new information with a concern of a very skewed sampling. However, it is worth noting that the authors have employed some statistical approaches to accommodate the skewedness. Additionally, the authors withheld a lot of information, especially in the introduction and methods. Finally, the study will add new information to the body of knowledge, and it is worth publishing if the authors address a few concerns raised below.

Introduction

Although the study is focused on the Indian “big four”, the introduction is scanty, just 273 words with only 3 references focusing on the main aims. The authors should consider introducing the nature of these variations as reported in other regions of the world, especially within Asia and within similar snake genera and species. This will inform the audience of the global aspect of potency and yield variations when it comes to snake venom.

Methods

The collection and measurement methods are so important in this study that it will be great if the authors add a stepwise protocol to the collection methods used and clarify how the collected venom was promptly measured before lyophilization. Additionally, the authors could clarify whether the same approach was used for the previously published study on snakes in captivity that was used for comparison.

Result

The results were well presented with clear figures. I will recommend that the authors consider adding “Supplementary Table 1: LD₅₀ values (mg/kg) of the ‘big four’ Indian snakes across biogeographic zones” as an adoption within the main manuscript since this is the main comparison of the potency which is part of the aim of the study. It should also contain information on the laboratory species in which the LD50 was determined.

Discussion

As mentioned earlier, it will be interesting for the audience to see how this Indian species differs or not compared with similar species found in other regions of the world especially in terms of potency and yield. Additionally, the authors focused more on the results than the implication of the results, it will be great if the authors will discuss further on how this new information will influence antivenom production, snakebite treatment, and the generally knowledge of these front fang snakes.

Reviewer #2: Overall Assessment

This manuscript presents a pan-India analysis of venom yields from wild “big four” snakes (Naja naja, Daboia russelii, Bungarus caeruleus, Echis carinatus), spanning six biogeographic zones and 338 milking events. It documents inter/intraspecific and ontogenetic differences and contrasts wild yields with captive reports. The dataset is valuable and clinically relevant.

Reviewer #3: Major findings.

1. In the methodology section, the authors should clearly indicate the parameters used to classify the individuals as neonate, juvenile, subadult and adult.

2. Results must be complemented with, at least, the HPLC/electrophoretic profile, biochemical, enzymatic and in-vivo analysis (proteomics also may be useful). The venom yielded by itself (and alone) is not worthy to be publisehd as a full research article.

3. The standard deviation (SD) is not a suitable parameter to indicate intraspecific variability.

4. In the discussion section, statements should be supported with the research data.

5. In the discussion section the authors are refereeing to results not mentioned or obtained in their research.

Minor findings.

1. Figure 1A. Please offer details about the approximately collecting areas. Where was collected each snake? India is such a huge and geographically diverse country and each of the showed biogeographic may include different ecogeographic compositions. Additionally, use colors to indicate which species was collected over each biogeographic area.

2. The statement on the paragraph 3 of the introduction is missing one important study about the big four venom variability. Senji Laxame also did a similar research in 2019. PLoS Negl Trop Dis 13(12): e0007899. https://doi.org/10.1371/journal.pntd.0007899.

3. On table 1, “NA” means no specimens were collected? If so, please just add 0 (zero) in the table. The letters NA are not suitable for quantification purposes.

4. Table 2 vs table 1 shows a difference on the number of D. russelii and B. caeruleus individuals collected and analyzed. Please correct.

5. On table 2 please decide to use one or two significative figures. After deciding, unify with the text.

6. On figure 4, “A” does not refer to “total yield” as stated in the figure foot note. It corresponds to the variation among the geographical region. B and C also do not correspond to what is shown in the figure. Finally, there is not figure D. Please correct.

7. Figure 2B is not mentioned in the text. It must be.

8. The text does not follow an order mentioning the figures. It goes from figure 2A to figure 4A. What happened (there is not mention of) with figures 2B, 2D, 3A, 3B and 3C? These kinds of gaps are not allowed according to the manuscript instructions of the PNTD (“Place figure captions in the manuscript text in read order”). https://journals.plos.org/plosntds/s/figures

9. The same as above is observed, in the same paragraph, jumping from figure 4A to figure 4B, the go back to figure 3A, move forward to figure 6A and finally go back to figure 2C and D. All in the same paragraph.

Reviewer #4: The manuscript reports a comprehensive study of venom yields in India's "big four" medically important snakes (Naja naja, Daboia russelii, Bungarus caeruleus, and Echis carinatus). By analyzing dry venom yields from 338 wild-caught specimens, the authors found substantial variation between species. N. naja and D. russelii have the highest yields. Geographic differences in intra-species venom yield were significant only in B. caeruleus, between the Gangetic Plains and Western Ghats. The life stage of the snakes was found to be the major difference, with adults producing more venom than juveniles or subadults. Comparison with captive populations showed that captivity does not significantly alter venom yield.

This study addressed a significant gap in snakebit epidemiology and snake venom biology in India as it appears to be one of the most comprehensive studies on venom yields of a very large number of wild-caught “big four” medically important snakes. Given the burden of snakebite envenoming in India, the value and impact of this study cannot be understated.

I do not have any major concerns or comments, other than a few minor points as followed:

1. Since the major difference in venom yield was found to be life stages, I have to wonder if (intra-species) body size difference has been controlled for? As body weight/length was not among the reported data, I have to assume venom yield is not adjusted for body size. Can the authors comment (also in the manuscript) if body size is a confounding factor in when comparing juvenile/subadult vs adult snake venom yield?

2. While sample numbers for each category of factors can be found in Supplementary Table 2, it would be great if the information can also be provided in the graphs itself.

3. The last paragraph in page 5 and the first paragraph in page 6 (immediately below Table 1 legend) seems repetitive.

4. Discussion: the second last paragraph discussed the correlations between venom yield with LD50. Obviously one of the major factors will be venom composition and the type of toxins present in the venom, Ie. common krait venom being predominantly neurotoxic will register a lower LD50 than the more hemotoxic venom of viper. Envenoming clinical presentation is also more complex than what LD50 can represent. Although the authors correctly point out there are multiple factors involved, can the authors expand this part of the discussion a little to present a more complete picture for the benefit of readers outside of the field.

Reviewer #5: (No Response)

PLOS authors have the option to publish the peer review history of their article (what does this mean? ). If published, this will include your full peer review and any attached files.

**Do you want your identity to be public for this peer review?** For information about this choice, including consent withdrawal, please see our Privacy Policy .

Reviewer #1: No

Reviewer #2: No

Reviewer #3: No

Reviewer #4: No

Reviewer #5: No

**Figure resubmission:**
---

## [Editor Report · Decision Letter 1]

22 Oct 2025

Dear Dr. Sunagar,

We are pleased to inform you that your manuscript 'Hiss and tell: What influences venom yields of India's big four snakes?' has been provisionally accepted for publication in PLOS Neglected Tropical Diseases.

Best regards,

Wuelton Monteiro, Ph.D.

Section Editor

Wuelton Monteiro

Section Editor

Shaden Kamhawi

co-Editor-in-Chief

Paul Brindley

co-Editor-in-Chief

---

## [Editor Report · Acceptance letter]

Dear Dr. Sunagar,

We are delighted to inform you that your manuscript, "Hiss and tell: What influences venom yields of India's big four snakes?," has been formally accepted for publication in PLOS Neglected Tropical Diseases.

Best regards,

Shaden Kamhawi

co-Editor-in-Chief

Paul Brindley

co-Editor-in-Chief
